# Development of Self-Administered Formulation to Improve the Bioavailability of Leuprorelin Acetate

**DOI:** 10.3390/pharmaceutics14040785

**Published:** 2022-04-03

**Authors:** Akie Okada, Rina Niki, Yutaka Inoue, Junki Tomita, Hiroaki Todo, Shoko Itakura, Kenji Sugibayashi

**Affiliations:** 1Faculty of Pharmacy and Pharmaceutical Sciences, Josai University, 1-1 Keyakidai, Sakado 350-0295, Saitama, Japan; gkd1902@josai.ac.jp (A.O.); gkm2118@josai.ac.jp (R.N.); yinoue@josai.ac.jp (Y.I.); sitakura@josai.ac.jp (S.I.); sugib@josai.ac.jp (K.S.); 2Research Analysis Center, Josai University, 1-1 Keyakidai, Sakado 350-0295, Saitama, Japan; jutomita@stf.josai.ac.jp

**Keywords:** lipid-based formulation, peptide drug, self-administered injection, leuprorelin acetate, micelles

## Abstract

In recent years, the development of self-injectable formulations has attracted much attention, and the development of formulations to control pharmacokinetics, as well as drug release and migration in the skin, has become an active research area. In the present study, the development of a lipid-based depot formulation containing leuprorelin acetate (LA) as an easily metabolizable drug in the skin was prepared with a novel non-lamellar liquid-crystal-forming lipid of mono-*O*-(5,9,13-trimethyl-4-tetradecenyl) glycerol ester (MGE). Small-angle X-ray scattering, cryo-transmission electron microscopy, and nuclear magnetic resonance observations showed that the MGE-containing formulations had a face-centered cubic packed micellar structure. In addition, the bioavailability (*BA*) of LA after subcutaneous injection was significantly improved with the MGE-containing formulation compared with the administration of LA solution. Notably, higher *C*_max_ and faster *T*_max_ were obtained with the MGE-containing formulation, and the *BA* increased with increasing MGE content in the formulation, suggesting that LA migration into the systemic circulation and its stability might be enhanced by MGE. These results may support the development of self-administered formulations of peptide drugs as well as nucleic acids, which are easily metabolized in the skin.

## 1. Introduction

Recently, many middle- and high-molecular-weight peptides have been approved for use because they show potent pharmacological effects in small doses. There is extensive ongoing research into the development of drug delivery methods for such peptides [1,2,3,4]; however, intravenous (i.v.) and subcutaneous (s.c.) routes are preferred because of their systemic effects due to the low membrane permeability of peptides. Drug delivery with s.c. injection has several advantages against i.v. administration because it can offer self-administration, reduced treatment burden, and lower health care costs [5,6,7]. In addition, i.v. drug administration provides the maximum plasma concentration (*C*_max_) just after administration, whereas a slower time to maximum plasma concentration (*T*_max_) and lower *C*_max_ are observed after s.c. injection due to slow absorption. These characteristics of s.c. injection might result in fewer hypersensitivity and infusion-related reactions than i.v. injection when a monoclonal antibody is administered [8]. However, incomplete bioavailability (*BA*) of biotherapeutic agents was observed after s.c. administration, for example leuprolide acetate (LA, *M.W.* 1.3 kDa) was unstable in the skin due to hydrolysis by enzymes and had a low absolute *BA* [9]. Ito et al. reported that microneedles based on sodium chondroitin sulfate improved skin stability by protecting LA degradation, resulting in increased the relative bioavailability [10]. Rahimi et al. have reported that improvement of LA stability by making complex with ß-cyclodextrin at various pH values (2.0–7.4) [11]. In addition, incomplete *BA* after s.c. injection might be related to the composition, volume, pH, and viscosity of the administered formulation [12].

Improvements in formulations may allow a lower frequency of s.c. injection by increasing the drug half-life and physicochemical stability in the administration site. Lipid-based liposomal formulations have attracted attention as a tool for depot formulations because they can encapsulate both hydrophilic and lipophilic drugs and show good biocompatibility, biodegradability, and low toxicity [13]. To increase the stabilization of liposomal formulations, Pluronic, linear nonionic triblock copolymers comprising polyethylene oxide (PEO) and poly-propylene oxide (PPO) is broadly used [14,15]. In addition, several reports have been published that formulations composed of phospholipid and Pluronic may improve the *BA* of hydrophobic compounds [16,17,18]. Furthermore, Shriky et al. [19] reported that injectable gel formulation consisted of Pluronic^®^ F-127 for controlled drug delivery. 

Recently, mono-*O*-(5,9,13-trimethyl-4-tetradecenyl) glycerol ester (MGE), an amphiphilic material, has gained attention as a novel additive to improve drug absorption. A formulation containing MGE enhanced direct nose-to-brain migration of an entrapped drug [20] and increased the transdermal permeation of a hydrophilic drug by altering membrane fluidity [21]. Therefore, the incorporation of MGE in a vehicle might be effective in improving the *BA* of middle and high molecular weight peptides by enhancing their migration into the systemic circulation.

In the present study, leuprorelin acetate (LA) was selected as a model middle-molecular-weight peptide that is easily metabolizable in the skin, and MGE-containing vehicles were prepared with Pluronic^®^ F-127 and phospholipids to improve the *BA* of leuprorelin acetate. As a phospholipid, 1,2-dioleoyl-*sn*-glycero-3-phosphoglycerol, sodium salt (DOPG), and 1,2-dioleoyl-*sn*-glycero-3-phosphocholine (DOPC) were selected to enhance entrapment efficacy of negatively charged LA with a positively charged vehicle.

## 2. Materials and Methods

### 2.1. Materials

MGE was kindly provided by Farnex Incorporated (Yokohama, Japan). LA was selected as the model drug and purchased from Japan Pharma Co., Ltd., (Tokyo, Japan). DOPG and DOPC were selected as unsaturated phospholipids and purchased from NOF Corporation (Tokyo, Japan). Pluronic^®^ F-127, nonionic surfactant, was purchased from Sigma-Aldrich (St. Louis, MO, USA). Other reagents and solvents were of special grade. The structural formulae of MGE, LA, and various phospholipids are shown in Table 1.

### 2.2. Methods

#### 2.2.1. Preparation of Lipid Particles

DOPC and DOPG with a molar ratio of 25:75 at 1.0 mM total lipid concentration were dissolved in ethanol in a flask, then evaporated at 50 °C to obtain a thin lipid layer. This ratio was chosen based on a preliminary study with different molar ratios of DOPC and DOPG (100:0, 75:25, 55: 45, 25:75) because it gave a small particle size with a small polydispersity index value (PDI) (Appendix A). Next, MGE: Pluronic^®^ F-127: DOPG was dissolved in ethanol at a molar ratio of 1:1:1 at total lipid concentrations of 1.0, 5.0, and 10 mM. After evaporation of ethanol, LA solution, prepared by dissolving in pH 7.4 phosphate-buffered saline (PBS) at 1.0 mg/mL concentration was added, followed by ultrasonic (VC-505, Sonics & Materials, Inc., Newtown, CT, USA) treatment for 30 s. This ultrasonic application process was repeated three times to obtain lipid suspensions. Finally, the lipid layer composed of DOPC and DOPG was hydrated with the lipid suspensions, followed by three ultrasonic treatments for 30 s each to obtain lipid formulations.

As a comparison, LA-entrapping liposomes were prepared by hydration of a thin lipid layer composed of DOPC and DOPG at a molar ratio of 25:75 with 1.0 mg/mL of LA solution. In addition, a lipid formulation with Pluronic^®^ F-127: DOPG at a molar ratio 1:1, MGE-free formulation, was also prepared. Furthermore, an LA-free formulation (blank formulation) was prepared with the same procedure without the addition of LA. Table 2 shows the formulation codes prepared in the present study. A physically mixed blank formulation was also prepared by mixing a blank formulation and 1 mM LA solution using a vortex mixer for 5 min at room temperature. Physically mixed formulations are indicated with Phy- at the beginning of the code name shown in Table 2. Figure 1 shows a schematic diagram of the preparation procedure of lipid-based self-administrated formulations.

#### 2.2.2. Apparent Entrapment Efficacy

The entrapment efficacy (*EE*) of LA in the prepared formulation except for liposomal ones (Lipo_1.0_ and Lipo_10_) was determined using an ultracentrifugation technique. The total LA content in the formulation (*C*_total_) was measured after disruption with acetonitrile, then centrifugation at 21,500× *g* for 5 min at 4 °C. Further centrifugation (21,500× *g* for 5 min at 4 °C) was done after mixing the obtained supernatant and PBS at a 1:1 ratio. LA-containing formulations, except for liposomal formulations, were filtered with Amicon Ultra 3k (Merck Millipore Ltd., MA, USA). Then LA concentration in the supernatant was measured to detect unentrapped LA in the formulation (*C*_out_). In case of liposomal formulations, the LA concentration in the obtained supernatant after centrifugation at 289,000 g for 60 min at 4 °C (Himac CS120GXII, Hitachi Koki Co., Ltd., Tokyo, Japan) was used to calculate *C*_out_. LA content was determined by liquid chromatography-tandem mass spectrometry (LC-MS/MS) to calculate %*EE*. %*EE* was calculated by the following equation: %*EE* = (*C*_total_ − *C*_out_)/*C*_total_ × 100.

#### 2.2.3. Physicochemical Characteristics of Prepared Formulations

Particle size and zeta-potential of the prepared formulations were measured using a Zetasizer Nano ZS (ZEN3600, Malvern Instruments Ltd., Malvern, Worcestershire, UK) using dynamic light scattering and electrophoretic light scattering, respectively. PDI was also calculated using Zetasizer software from the obtained size distribution of the prepared formulation. The prepared formulation was diluted 10 times with purified water to prepare the measurement sample, and then the measurement was carried out.

#### 2.2.4. Small-Angle X-ray Analysis

The liquid structure was analyzed using a small-angle X-ray diffractometer (SAXSpace (Anton Paar, Graz, Austria) equipped with a one-dimensional detector (Mythen R 1k) using Cu Kα radiation (λ = 0.154 nm) with an accelerating voltage of 50 kV and an applied current of 40 mA. The samples were measured in line collimation mode using a TCStage 150 with quartz capillary for a 2 h exposure period.

#### 2.2.5. Observation with a Cryo-Transmission Electron Microscope 

The formulation was observed using a cryo-transmission electron microscope (Cryo-TEM) (JEM-3100FEF, JEOL Ltd., Akishima, Tokyo, Japan). For imaging, 1 μL of a 20-fold dilution of the formulation was dropped onto a hydrophilized copper grid (200 mesh, JEOL Corporation) and blotted. The samples were rapidly frozen using ethane as a freezing solvent with a rapid freezing system (EM-CPC, Leica Microsystems Japan, Tokyo, Japan) for observation using the Cryo-TEM at 5–10 μm defocus.

#### 2.2.6. H-^1^H Nuclear Overhauser Effect Spectroscopy (NOESY) Nuclear Magnetic Resonance Spectroscopy

The formulations were prepared using the same method as described in Section 2.2.1, but deuterium oxide (D_2_O) was used instead of distilled water with the same volume to prepare the formulations. In addition, formulation components, except for MGE due to the self-organization property of non-lamellar liquid crystal (NLLC) structure by contacting with water, were dissolved in D_2_O. Nuclear magnetic resonance (NMR) spectroscopy analysis was performed using an NMR spectrometer (Varian NMR System 700 MHz, Agilent Technologies, Inc., Santa Clara, CA, USA) equipped with a cold probe (1H{13C/15N} 5 mm Triple Resonance 13C Enhanced Cold Probe, VT, 700NB). The measurement was conducted using the following conditions: a pulse width of 10.05 µs and 10.75 µs for P and MP formulations, respectively, a relaxation delay of 1.0 s, an accumulation number of 16, a mixing time of 500 s, and a temperature of 298 K.

#### 2.2.7. Measurement of Viscosity

An electron-magnetically spinning viscometer [22,23,24] (EMS-100, Kyoto Electronics Manufacturing Co., Ltd., Kyoto, Japan) was used to measure the viscosity of prepared formulations. The viscometer has a couple of magnets attached to the rotator, which apply a rotating magnetic field. A test tube with a smooth concave bottom, in which the sample and 2-mm aluminum spheres were included, set at the center of the magnetic field. The measurement was conducted at 25 °C (*n* = 50). The viscosity was shown as the average value measured 50 times for each sample.

#### 2.2.8. In Vitro LA Release from the Prepared Formulations

The prepared formulation (100 µL) was added to a Pur-A-Lyzer TM mini 12,000 dialysis kit with a fractional molecular weight of 12,000 (Sigma Aldrich, St. Louis, MO, USA) and placed in a vial containing 40 mL PBS (1/30 M) with a stirrer bar. An in vitro release test was conducted for 6 h in a water bath at 37 ± 0.02 °C with a stirring speed of 100 rpm. At predetermined times, 500 µL of the solution was collected from the receiver side and the same volume of PBS was added to maintain a constant volume. The sample was mixed with the same volume of acetonitrile, then mixed using a vortex mixer for 5 min. Then, the obtained samples were stored at −80 °C until measurement.

#### 2.2.9. In Vivo Experiment 

Male Wistar rats (body weight 200 ± 20 g, 8-weeks old (Sankyo Labo Service Corporation, Inc., Tokyo, Japan)) were selected. The rats were housed in a room regulated at 25 ± 2 °C with a light/dark cycle (on, off time: 09:00–21:00) every 12 h. Water and feed (MF (Oriental Yeast Co., ltd., Tokyo, Japan)) were freely accessed. All procedures were approved by the Josai University Animal Care and Use Committee and complied with the National Institutes of Health’s Guide for the Care and Use of Laboratory Animals. After approval by the Josai University Ethics Committee (approved number: JU20005), the experimental animals were used in accordance with the Josai University Laboratory Animal Regulations. After the rats were anesthetized with triple anesthesia (intraperitoneal administration of 0.15 mg/kg medetomidine hydrochloride, 2.5 mg/kg butorphanol, 2 mg/kg midazolam tartrate), they were placed on their backs and the jugular vein and abdomen, which were the blood collection points, were shaved with clippers. LA solution or prepared formulation was administered subcutaneously into the rat abdomen using a 23 G needle (TERUMO Co., Tokyo, Japan) at an LA dose of 2.0 mg/kg. After blood collection, the same volume of saline solution was injected through the tail vein using a 27 G winged needle (TERUMO Co., Tokyo, Japan). The obtained blood sample was centrifuged (21,500× *g*, 5 min, 4 °C) to obtain plasma. The same amount of acetonitrile was added to the obtained plasma and mixed for 5 min with a vortex mixer, then centrifuged again (21,500× *g*, 5 min, 4 °C) to remove the protein, and the sample was stored at −80 °C until measurement. The area under the concentration–time curve until 12 h after administration (*AUC*_12h_), was calculated using the trapezoidal rule. Relative *BA* was calculated as the ratio of *AUC*_12h_ obtained from s.c. administration of each prepared formulation to that of the LA solution.

#### 2.2.10. Liquid Chromatography-Tandem Mass Spectrometry Condition to Detect LA

The LC-MS/MS system consisted of a system controller (CBM-20A; Shimadzu Corporation, Kyoto, Japan), pump (LC-20AD; Shimadzu Corporation), auto-sampler (SIL-20AC; Shimadzu Corporation), column oven (CTO-20AC; Shimadzu Corporation), detector (3200 QTRAP; AB Sciex, Tokyo, Japan), and analysis software (Analyst^®^ version 1.4.2; Shimadzu Corporation). The column and the guard column were Shodex^®^ ODP2 HP-2B 2.0 mm × 50 mm and ODP2 HPG-2A 2.0 mm × 10 mm, respectively (each from Showa Denko, Tokyo, Japan). The column temperature was adjusted to 40 °C. An internal standard method was used for the TA assay, with betamethasone valerate used for this purpose. A mixed solution (A:B, 70:30) was used for the mobile phase, where A was 0.1% formic acid purified water, and B was acetonitrile. The flow rate was 0.2 mL/min, and the injection volume was set to 10 μL. Electrospray ionization was used for LA ionization. The measured molecular weight of LA was set to *m*/*z* 605.30 for the precursor ion and *m*/*z* 249.00 for the product ion. The ion spray voltage was 5000 V, the nebulizer gas pressure was 80 psi, the drying gas flow rate was 10 L/min, and the drying gas temperature was 600 °C. The lower limit of quantification of this assay was 1 ng/mL.

#### 2.2.11. Statistical Analysis

Statistical analysis was performed using JMP^®^ Pro (ver. 15.0.0, SAS Institute Inc., Cary, NC, USA). The statistical significance of differences was examined using a one-way analysis of variance (ANOVA) followed by a Tukey–Kramer post hoc test. The significance level was set at *p* < 0.05. All experimental measurements were performed at least in triplicate.

## 3. Results

### 3.1. Characteristics of the Prepared Formulations

Figure 2 shows naked eye observation of the dispersion results of the prepared formulations. A highly transparent dispersion was observed for the prepared formulations. The size (a), zeta-potential (b), PDI (c), and %*EE* (d) of LA for the prepared formulations are shown in Figure 3. The particle size decreased with the increase in MGE concentration in the MP formulations. In addition, MP_10_ displayed a small PDI compared with the other formulations. On the other hand, Lipo formulations exhibited a large particle size compared with MP and P formulations, the mean particle sizes for Lipo_1.0_ and Lipo_10_ were 250 nm and 320 nm, respectively. Although the zeta-potential of all prepared formulations exhibited negative values, especially Lipo formulations showed a high negative zeta potential.

The %*EE* of LA in the MP and P formulations showed a higher value (above 79%*EE*), although Lipo formulations exhibited lower LA contents (Lipo1.0 and Lipo10 were 22.3%*EE* and 66.9%*EE*, respectively). The viscosity of the prepared formulations of LA solution, MP_10_, and P_10_ were 9.4 × 10^−1^, 2.0 × 10^1^, and 1.4 × 10^1^ mPa·s, respectively, which were suitable for an injectable viscosity with a needle. 

### 3.2. Cryo-TEM Observation

Figure 4 shows cryo-TEM observation results of MP_1.0_, MP_5.0_, MP_10_, P, and Lipo_10_. Micelle structures were confirmed from observations of MP_1.0_, MP_5.0_, MP_10_, and P. On the other hand, Lipo_10_ displayed a unilamellar structure. Negative-stained cryo-TEM observations were performed with MP_5.0_, MP_10_, and P to confirm the internal structure. Clusters constructed with well-ordered micelles were confirmed in both formulations of MP_5.0_, MP_10_, and P from negative-stained cryo-TEM observations. The size of clusters and unilamellar particles observed in cryo-TEM observations almost corresponded with the particle size measured using a Zetasizer Nano ZS.

### 3.3. SAXS Analysis

Figure 5 shows the X-ray diffraction patterns of MP_1.0_, MP_5_, MP_10_, and P. Peaks were observed in formulations with higher concentrations of Pluronic^®^ F-127 (MP_5_, MP_10_, and P). A closely similar peak ratio with (√3: √4: √8) was observed in both MP_10_ and P, although the first peak of MP_5_ was broadened. According to the peak ratio, the structure of MP_10_ and P was identified as face-centered cubic (FCC). Together with cryo-TEM observation results, MP_5_, MP_10_, and P constructed FCC-packed micelles.

### 3.4. Nuclear Magnetic Resonance

{1H-1H} NOESY NMR spectroscopy was performed to investigate the interactions between the formulation components of P (Figure 6a,b) and MP_10_ (Figure 6c,d). A comparison of the obtained NOESY NMR spectra of P and MP_10_ showed that an MGE-related peak appeared around 4.96 ppm only in MP_10_ (Figure 6c,d, *f*_2_). MGE showed cross-peaks with Pluronic^®^ F-127 (Figure 6c,d, *f*_1_ = 0.93 ppm), oleyl groups of DOPG and DOPC (Figure 6c,d, *f*_1_ = 1.50 ppm and 2.19 ppm, respectively), and LA (Figure 6c,d, *f*_1_ = 1.77 ppm), indicating the presence of interactions. The interaction of the methyl group of the PPO segment in Pluronic^®^ F-127 appeared at around 0.97 ppm (Figure 6, *f*_2_), and that of the methylene group (Figure 6, *f*_1_) observed at around 3.46 ppm was also confirmed.

### 3.5. In Vitro Release Results

Figure 7 shows the LA release profiles from the prepared formulations. The observed *Q* was higher in the following order; Lipo_1.0_ > MP_1.0_ > MP_5.0_ > MP_10_ ≅ P. Lipo formulations showed the 40 to 60% of LA release, whereas MP and P formulations exhibited less than 20% of *Q* value. Especially, MP formulations showed lower LA release as the lipid and Pluronic^®^ F-127 concentrations were increased. The LA release from the Phy-MP10 and Phy-P formulations was higher than that of the MP10 and P formulations.

### 3.6. In Vivo Experiment

Figure 8 shows the blood concentration profile of LA after s.c. injection. When LA solution was administered, the LA concentration increased, and *C*_max_ of 34.2 ng/mL was observed around 1.0 h after administration. An almost similar blood concentration profile of LA (*C*_max_ and *T*_max_) was observed as Lipo_10_ was administered. The highest LA concentration (*C*_max_: 98.5 ng/mL) was confirmed 4 h after the administration of MP_10_, whereas the other MP formulations showed only slightly higher *C*_max_ values. On the other hand, P_10_ showed delayed *T*_max_ (6 h) with slightly higher *C*_max_ compared with LA solution.

MP_1.0_, Lipo_1.0_, and Lipo_10_ formulations show similar absolute *BA* values compared with an LA solution, whereas about 11- and 7-fold higher absolute *BA* were observed with MP_10_ and P_10_, respectively, when a 23 G needle was used for administration.

The effect of encapsulation of LA in the formulation on the blood concentration profile of LA was also investigated with a physically mixed blank formulation and LA solution. Figure 7 shows the blood concentration profile of LA after s.c. administration of Phy-MP_10_ and Phy-P formulations (Figure 8b). When Phy-P was administered, the observed *C*_max_ value was almost the same, although a lower *T*_max_ value was confirmed when P_1_ was administered. On the other hand, lower *T*_max_ and *C*_max_ were observed when Phy-MP_10_ was administered compared with MP_10_. MP_10_ exhibited an approximately 2-fold higher blood concentration of LA and *BA* than Phy-MP_10_ (Table 3).

## 4. Discussion

LA-containing lipid dispersions incorporating NLLC forming lipids were prepared successfully. However, typical NLLC structures such as hexagonal and reverse hexagonal structures were not confirmed, and micelles with ordered structures were observed using cryo-TEM observation. Block copolymer such as Pluronic^®^ F-127 has been used to prepare micelles in nanomedicine applications [14,15]. Pluronic spontaneously forms micelles at concentrations equal to or above the critical micellar concentration (cmc; 0.8 wt%) [26,27,28]. The prepared formulation in the present study had a higher concentration than cmc (MP_1.0_:1.26 wt%, MP_5.0_: 6.30 wt%, MP_10_: 12.6 wt%, respectively, when dissolved in 10 mL), so the micellar structures were observed in cryo-TEM observations. 

NMR and SAXS results revealed that prepared MP formulations formed micellar structures, and SAXS observation results showed that FCC-packed micelles were obtained [29,30]. Pluronic^®^ F-127 has a triblock polymer with two hydrophilic tails in the structure. Interactions were observed between the methyl and methylene groups of the PPO segment in Pluronic^®^ F-127, suggesting the construction of micellar structures [31], as illustrated in Figure 9. Cryo-TEM observation results also supported the particle structure. The first peak in the SAXS diffraction pattern for MP_1.0_ was, however, not obviously observed in the current experiment condition. According to cryo-TEM observation result in MP_1.0_, the number of constructed particles was less than MP_5.0_ and MP_10_ formulations that had higher lipid concentrations. Thus, increasing the total lipid concentration in the formulation would be related to the construction of a face-centered cubic structure in the present study, which was also confirmed by cryo-TEM observation results.

LA absorption from P after s.c. injection displayed an improved *BA* compared to LA solution and Lipo formulations. Because P formed a micellar structure, a long length of hydrophilic chain derived from Pluronic^®^ F-127 has the potential to improve the *BA* by reducing enzymatic degradation [32]. The hydrophilic block of poly (ethylene oxide) in Pluronic^®^ F-127 can form hydrogen bonds with the aqueous surroundings and form a tight shell around the micellar core. The micelles with PEO corona could resist protein adsorption. As a result, the structure is effectively protected against enzymatic degradation and hydrolysis. Physically mixed formulations composed of LA solution and blank P (Phy-P) were administered via s.c. injection and decreased *BA* was observed compared with s.c. administration of P.

Cervin et al. reported that the *BA* of somatostatin was improved after intravenous administration by the protective effect of encapsulation into lipid-based liquid crystalline nanoparticles from enzymatic degradation [33]. Therefore, our results also suggested that LA entrapped in the micelles may be helpful to improve the stability at the injection site. Even though LA was distributed in the external phase of micelles, higher *BA* was confirmed in Phy-P compared with LA. Thus, the interaction of LA with a long length of hydrophilic block tail of PEO, the hydrophilic moieties in Pluronic^®^ F-127 might happen to hinder enzymatic degradation. The same tendency was also observed when Phy-MP_10_ was administrated. On the other hand, *BA* values obtained from the administration of Lipo formulations and MP_1.0_ were similar to that of the LA solution. For the Lipo formulations, positively charged LA may be located in the negatively charged hydrophilic region, the external phase of liposomes. The presence of LA in the external region of the prepared formulations also occurred in physically mixed formulations. However, LA release from the Lipo formulations was faster than from other formulations, except for the LA solution. Therefore, a weak interaction might be the reason for the lower *BA* provided by Lipo formulations. In the case of MP_1.0_, LA was located in the internal and external phases of micelles. The Pluronic concentration in MP_1.0_ (1.26 wt%) was lower than in MP_10_ and P formulations (12.6 wt%, respectively). Thus, the complexity of the external phase structure induced by Pluronic might be related to LA stability. Figure 9 shows a schematic representation of the morphology of the prepared formulations illustrated from the NMR observation. 

MP_10_ exhibited higher *C*_max_ and significantly improved *BA* compared with other formulations. The obtained *BA* increased with an increase in MGE content in the formulation; 4.0% for MP_1.0_, 12.7% for MP_5.0_, 44.8% for MP_10_. Libster et al. reported that hydrophilic proteins interact with the polar moieties of glycerin monooleate, an NLLC structure forming lipid [18]; this interaction may improve the thermal stability of the formulation. Furthermore, increased lipophilicity caused by the interaction might be effective for increasing membrane permeation. In a study of antimicrobial activity, a ß-turn conformation of the peptide was induced by the addition of micelles [34]. Moreover, rapid permeation via a transcellular route was confirmed with a peptide that formed a ß-turn conformation due to the greater lipophilic properties. In the present study, the interaction of LA with MGE was also confirmed by NMR observation. Several reports have been published that improved drug permeation by MGE was caused by increased membrane fluidity [21]. Furthermore, enhancement of the effect by MGE was higher in forming an unorganized state than in constructing an NLLC structure [21]. Thus, an increase in the lipophilicity provided by a weak interaction with the hydrophilic moieties of MGE and LA and the permeation enhancement effect by MGE might be reasons for the improvement of blood transfer from the injection site. In addition, regarding the effect of constructed MP formulation forms, a crystalline state on the bioavailability of LA was unrevealed. Further experiments should be done to reveal the constructed formulation forms and improved BA of LA. 

## 5. Conclusions

In the present study, micelles incorporating MGE significantly improved the *BA* of LA, suggesting that MGE would be a useful additive to injectable formulation to increase the utilization of LA after s.c. administration. Although further experiments should be performed to reveal the usefulness of micelles containing MGE through investigations with other middle-molecular-weight compounds, this result may contribute to the development of self-injectable formulations.

## Figures and Tables

**Figure 1 pharmaceutics-14-00785-f001:**
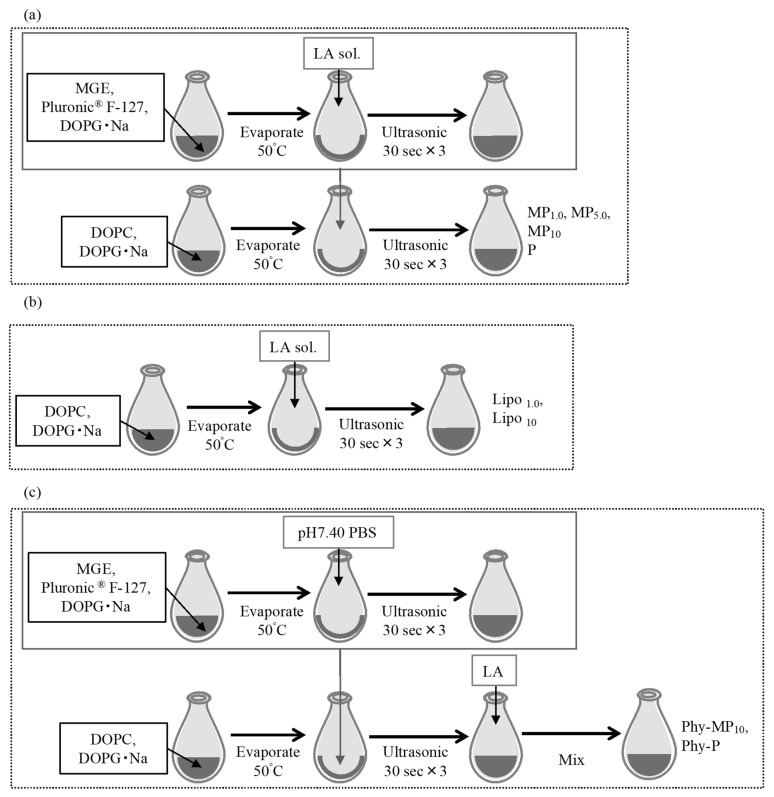
Schematic diagram of the preparation procedure of lipid-based self-administrated formulations. (**a**) MP and P formulations, (**b**) Lipo formulations, (**c**) Phy-MP_10_ and Phy-P.

**Figure 2 pharmaceutics-14-00785-f002:**
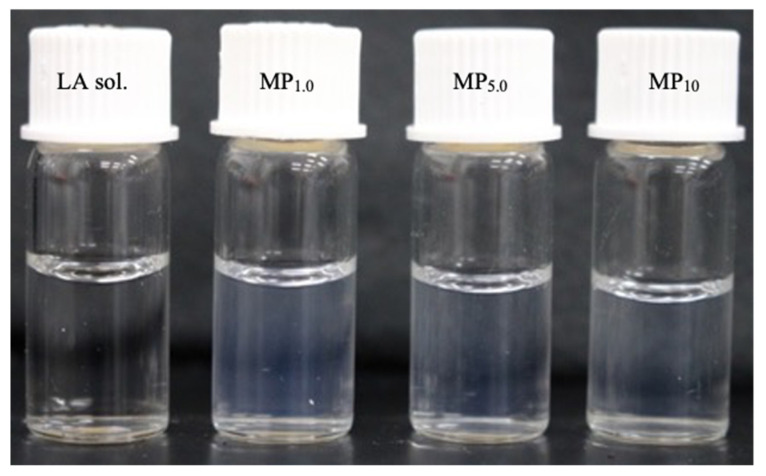
Visible observations of prepared formulations with the naked eye.

**Figure 3 pharmaceutics-14-00785-f003:**
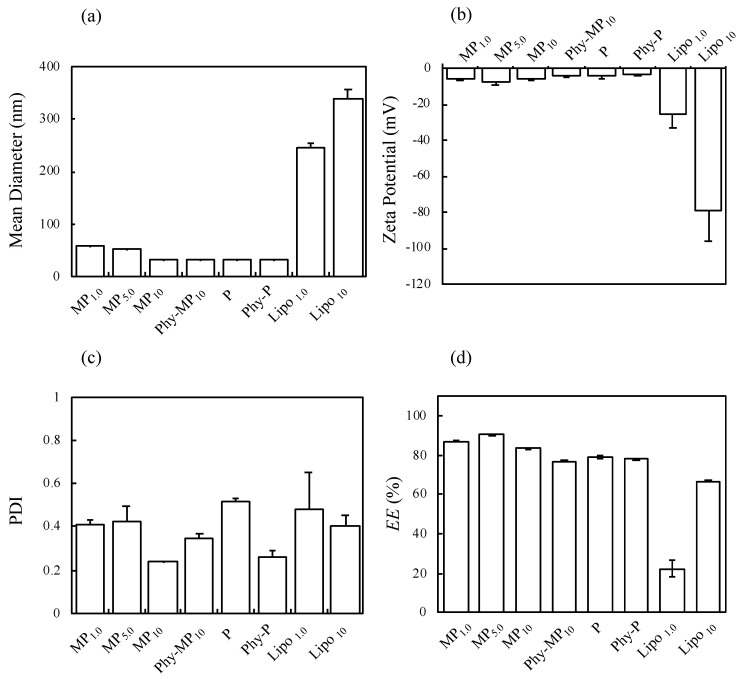
Measurement of particle size (**a**), zeta-potential (**b**), polydispersity index (**c**), and (**d**) entrapment efficacy of LA for prepared formulations. Each value shows the mean ± S.D. (*n* = 3).

**Figure 4 pharmaceutics-14-00785-f004:**
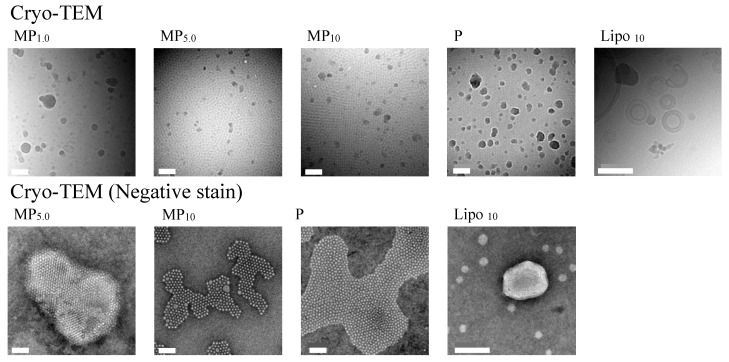
Cryo-TEM and negative-stained cryo-TEM observation results. The white bar indicates 100 nm.

**Figure 5 pharmaceutics-14-00785-f005:**
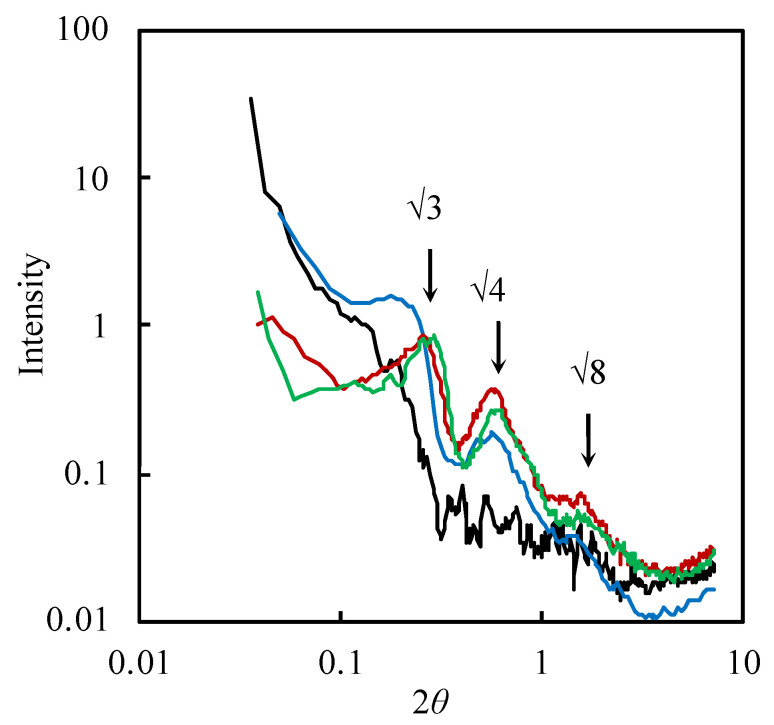
Small-angle X-ray scattering observation results of the MP and P formulations. Black line: MP_1.0_, blue line: MP_5.0_, red line: MP_10_, and green line: P.

**Figure 6 pharmaceutics-14-00785-f006:**
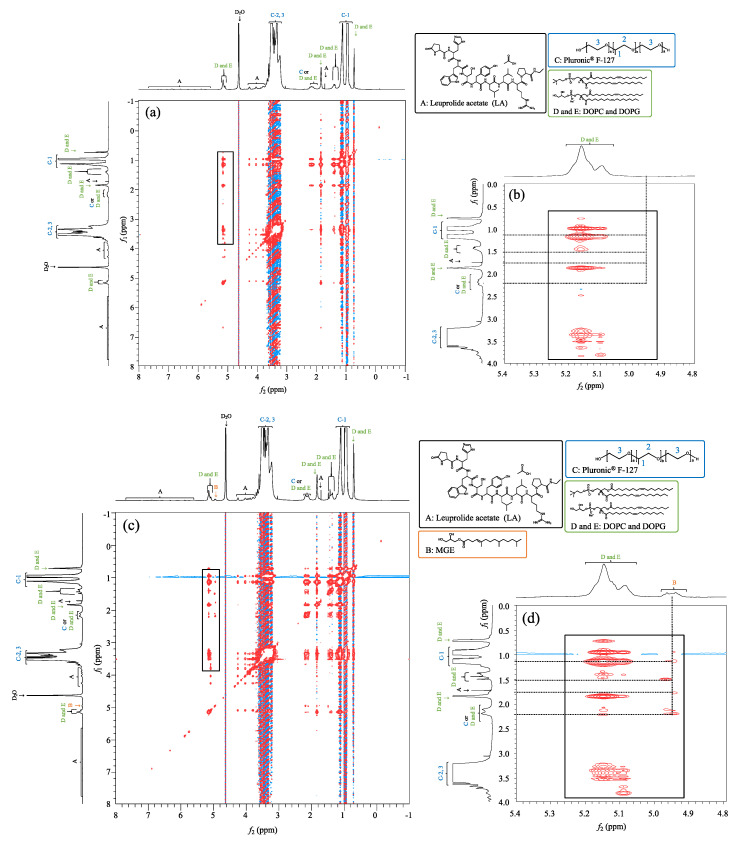
{1H-1H} NOESY NMR spectroscopy of P (**a**,**b**) and MP_10_ (**c**,**d**). (**b**,**d**): enlarged areas of the black boxes in (**a**,**c**), respectively.

**Figure 7 pharmaceutics-14-00785-f007:**
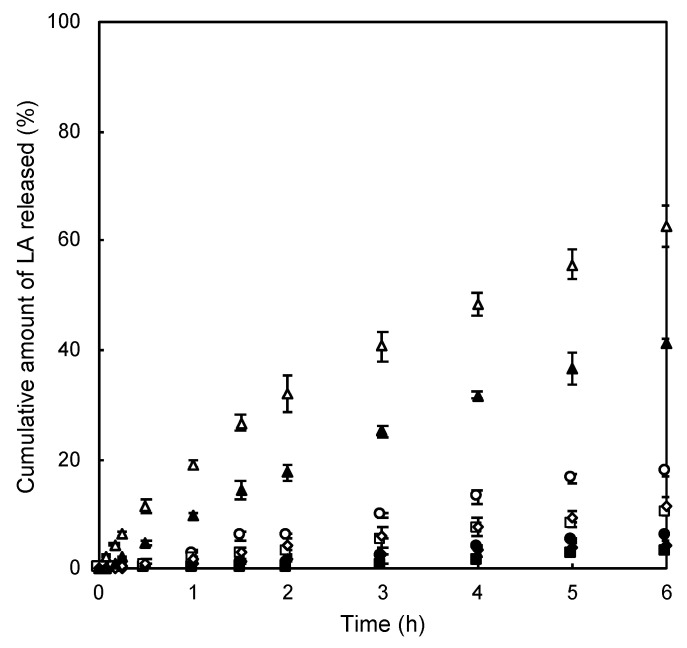
LA release profile from each formulation against the time as the X-axis. Symbols; O: MP_1.0_, ●: MP_5.0_, ■: MP_10_, □: Phy-MP_10_, ♦: P, ◊: Phy-P, Δ: Lipo_1.0_, ▲: Lipo_10_. Each value shows the mean ± S.E. (*n* = 3–5).

**Figure 8 pharmaceutics-14-00785-f008:**
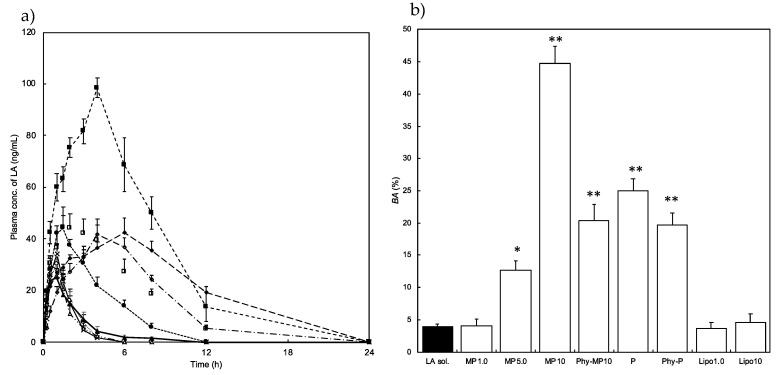
Plasma concentration profile (**a**) and bioavailability (**b**) of LA after subcutaneous injection of the prepared formulations. (**b**) Y-axis enlargement in (**a**). Symbols; ×: LA solution, O: MP_1.0_, ●: MP_5.0_, ■: MP_10_, □: Phy-MP_10_, ♦: P, ◊: Phy-P, Δ: Lipo_1.0_, ▲: Lipo_10_. Each value shows the mean + S.E. (*n* = 3). (**b**) * *p* < 0.05 compared with LA sol, ** *p* < 0.01 compared with LA sol.

**Figure 9 pharmaceutics-14-00785-f009:**
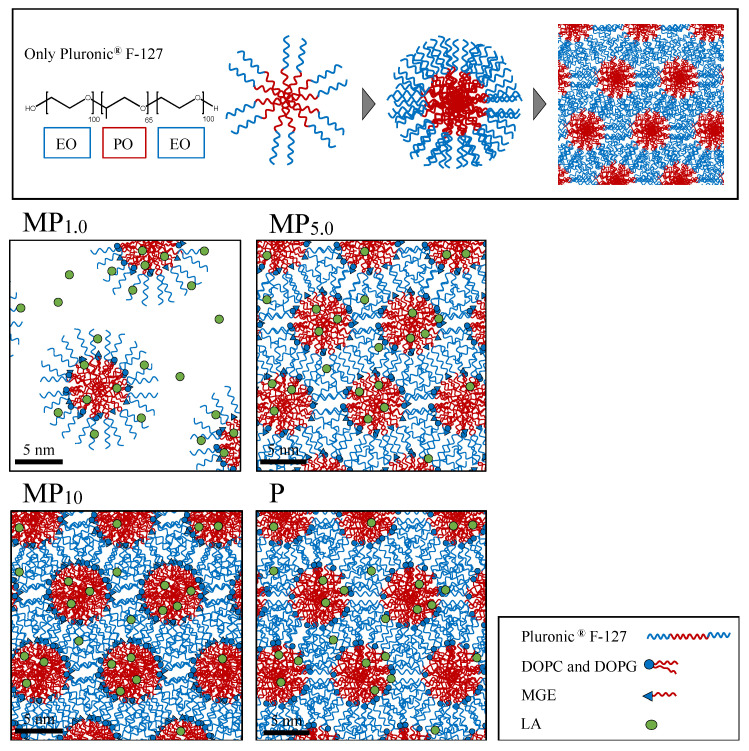
Schematic representation of the morphology of the prepared formulations.

**Table 1 pharmaceutics-14-00785-t001:** Structure and physicochemical properties of drug and formulation components used in the present study.

	Structure	*M.W.*	*XLogP_3_*	*PI*
Leuprolide acetate (LA)	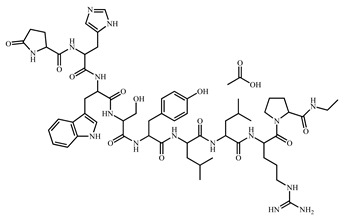	1269.5	-	9.1
mono-*O*-(5,9,13-trimethyl-4-tetradecenyl) glycerol ester (MGE)	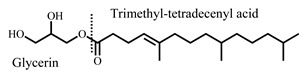	470.7	-	-
Pluronic^®^ F-127	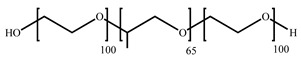	~12,600	-	-
1,2-Dioleoyl-*sn*-glycero-3-phosphocholine (DOPC)	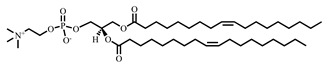	786.1	12.8	-
1,2-Dioleoyl-*sn*-glycero-3-phosphoglycerol, sodium salt(DOPG · Na)	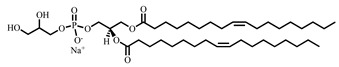	797.0	12.5	-

**Table 2 pharmaceutics-14-00785-t002:** Composition of the prepared formulations.

mM	LA sol.	MP_1.0_	MP_5.0_	MP_10_	P	Lipo_1.0_	Lipo_10_
MGE	-	1	5	10	-	-	-
Pluronic^®^ F-127	-	1	5	10	10	-	-
DOPG	-	1.75	5.75	10.75	10.75	0.75	7.5
DOPC	-	0.25	0.25	0.25	0.25	0.25	2.5

**Table 3 pharmaceutics-14-00785-t003:** Calculated AUC after i.v. or s.c. administration of formulation.

Formulation	Route	AUC_0–24 h_ (ng · h/mL)
LA sol. ^#^	i.v.	3816 ± 458
LA sol.	s.c.	59.7 ± 6.49
MP_1.0_	s.c.	61.1 ± 16.0
MP_5.0_	s.c.	193 ± 23.0
MP_10_	s.c.	683 ± 41.1

^#^ The *BA* value after i.v. injection of LA sol. (5 mg/kg ) was calculated by the blood concentration-time profile until 6 h after the administration with a trapezoidal method [25].

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
