# Peer review of "Development of Self-Administered Formulation to Improve the Bioavailability of Leuprorelin Acetate"

_pharmaceutics, 2022, doi:10.3390/pharmaceutics14040785_

Round 1

Reviewer 1 Report

In this study, the authors developed a lipid-based depot formulation containing leuprorelin acetate (LA) as a metabolizable drug model in the skin. A full physicochemical characterization of the formulation was performed that was then tested in vivo. The study is interesting and sufficiently innovative, showing a practical approach for the development of a novel lipidic formulation. Experimental plan has been accurately designed and performed  and results have been exhaustively discussed. Conclusion is supported by the results. The scientific soundness is of high level. I recommend the acceptance of this study in the present form.

Author Response

Thank you for your comment. I have revised several points that pointed out by another reviewers. I hope that you will agree with the revised points.

Reviewer 2 Report

In the present study, the development of a lipid-based depot formulation with a non-lamellar liquid-crystal-forming lipid of mono-O-(5,9,13- 16 trimethyl-4-tetradecenyl) glycerol ester (MGE) is presented.

Authors present the methods of preparation, in-vitro release as well as in vivo profiles. Also, very extensive investigation on “crystalline” structure is presented.

Following aspects, I would consider as not sufficient or “wrongly” presented/described.

  • Release only up to 6 h. However, highest release only 60 %. Linear relationship release vs. square root of time is without comments?
  • 8 C. Values for MP10 and P are marked with a star without explanation.
  • 8. No data between 12 and 24 h and just connected with straight line. Therefore, the AUC0-24 h is calculated with significant approximation. I would suggest to present AUC0-12 h or connecting 12 and 24 h by extrapolation (kel must be known from iv data, which, however not presented).
  • Table 3 presents AUC values, according to which the highest BA (%) is approx. 20 % for formulation. However, in Fig. 8c, the highest BA (%) is nearly 50 %. This problem is true for BA of all other formulations.
  • In the discussion section, extensive discussion about “crystalline” state, but not connected with discussion about bioavailability.
  • The writing style is, in principle, clear but very unwieldy.

Author Response

In the present study, the development of a lipid-based depot formulation with a non-lamellar liquid-crystal-forming lipid of mono-O-(5,9,13- 16 trimethyl-4-tetradecenyl) glycerol ester (MGE) is presented.

Authors present the methods of preparation, in-vitro release as well as in vivo profiles. Also, very extensive investigation on “crystalline” structure is presented.

Following aspects, I would consider as not sufficient or “wrongly” presented/described.

Q1. Release only up to 6 h. However, highest release only 60 %. Linear relationship release vs. square root of time is without comments?

A1. Thank you for your comment. According to your comment, we removed the figure of linear relationship release vs. square root of time from Fig.7. In addition, more detail explanation was added in the result section.

Figure 7 shows the LA release profiles from the prepared formulations. The observed Q was higher in the following order; Lipo1.0 > MP1.0 > MP5.0 > MP10 @ P. Lipo formulations showed the 40 to 60% of LA release, whereas MP and P formulations exhibited less than 20% of Q value. Especially, MP formulations showed lower LA release as the lipid and Pluronic® F-127 concentrations were increased. The LA release from the Phy-MP10 and Phy-P formulations was higher than that of the MP10 and P formulations.

Q2. Fig. 8C Values for MP10 and P are marked with a star without explanation.

A2. Thank you for your comment. We added the explanation in the figure caption.

Q3. No data between 12 and 24 h and just connected with straight line. Therefore, the AUC0-24 h is calculated with significant approximation. I would suggest to present AUC0-12 h or connecting 12 and 24 h by extrapolation (kel must be known from iv data, which, however not presented).

A3. Thank you for your comment. According to your comment, the AUC0-12h was listed in the Table 3 after the recalculation.

Q4. Table 3 presents AUC values, according to which the highest BA (%) is approx. 20 % for formulation. However, in Fig. 8c, the highest BA (%) is nearly 50 %. This problem is true for BA of all other formulations.

A4. Thank you for your comment. The injection dose of LA was different between intravenous and subcutaneous injections. The data ofi.v. injection was referred from our previous report. The following sentence was added in the cation of Table 3. In addition, our previous report was added as reference number of 25, so other reference numbers were changed accordingly.

#The BA value after i.v. injection of LA sol. (5 mg/kg ) was calculated by the blood concentration-time profile until 6 h after the administration with a trapezoidal method[25].

Q5. In the discussion section, extensive discussion about “crystalline” state, but not connected with discussion about bioavailability. The writing style is, in principle, clear but very unwieldy.

A5. Thank you for your comment. The pointed you out comment is very important part in this manuscript. However, with the current data, it would be difficult to reveal the relationship between the constructed structure of the prepared formulation and the improved bioavailability of LA. Thus, possible mechanisms that might not be directly connected with formed formulation were added in the discussion part.

The following sentences were added in the discussion part.

In addition, the effect of constructed MP formulation forms, crystalline state, on the bioavailability of LA was unrevealed. Further experiments should be done to reveal the constructed formulation forms and improved BA of LA.

Reviewer 3 Report

This is a well-organized paper, I suggest acceptance after the revision. The suggestions as follow: 

1) For section 2.2.2 Entrapment efficacy, could you please give me detailed information on how to calculate %EE using LC-MS/MS? 

2) For DLS testing, the formulation was diluted 10 times, why the sample was diluted 20 times for cryo-TEM? 

3) For section 3.1, line 243-244, except for viscosity value, is there any direct method to prove the suitable injectable viscosity with a needle for the prepared sample? 

4) For Figure 3, are there diameter distribution for each sample, I suggest such data could be provided in Supporting information. More, for zeta potential value, it seems that the sample for MP1.0, MP5.0, MP10, Phy-MP10, P, and Phy-P are not stable. 

5) the resolution of Figure 6 is low, please provide the improved one. 

6) For Figure 7 and Figure 8, please improve the quality of them, such as the symbols in both captions to indicate in figures were confused and not clear, and for Figure 8b, I suggest the author separate this figure into two, then the readers could understand clearly. 

7) How about the toxicity of such formulation on other organs, were there some data? 

Author Response

This is a well-organized paper, I suggest acceptance after the revision. The suggestions as follow:

Q1) For section 2.2.2 Entrapment efficacy, could you please give me detailed information on how to calculate %EE using LC-MS/MS?

A1. Thank you for your comment. We changed the methodology part as follows:

The entrapment efficacy (EE) of LA in the prepared formulation except for liposomal ones (Lipo1.0 and Lipo10) was determined using an ultracentrifugation technique. The total LA content in the formulation (Ctotal) was measured after disruption with acetonitrile, then centrifugation at 21,500 g for 5 min at 4°C. Further centrifugation (21,500 g for 5 min at 4°C) was done after mixing the obtained supernatant and PBS at a 1:1 ratio. LA-containing formulations except for liposomal formulations were filtered with Amicon Ultra 3k (Merck Millipore Ltd, MA, USA). Then LA concentration in the flowthrough was measured to detect unentrapped LA in the formulation (Cout). In case of liposomal formulations, the LA concentration in the obtained supernatant after centrifugation at 289,000 g for 60 min at 4°C (Himac CS120GXII, Hitachi Koki Co., Ltd., Tokyo, Japan) was used to calculate Cout. LA content was determined by liquid chromatography-tandem mass spectrometry (LC-MS/MS) to calculate %EE. %EE was calculated by the following equation: %EE =(Ctotal-Cout)/Ctotal Í100.

Q2) For DLS testing, the formulation was diluted 10 times, why the sample was diluted 20 times for cryo-TEM?

A2) Thank you for your comment. As you questioned, it might be better to conduct the experiments with the same dilute magnification. Cryo-TEM observation was done under condition that was suitable to observe. Thus, the dilute magnification was different.

Q3) For section 3.1, line 243-244, except for viscosity value, is there any direct method to prove the suitable injectable viscosity with a needle for the prepared sample?

A3) Thank you for your comment. We also considered a measurement method to determine the viscosity of the prepared formulations with a needle. However, the size and design of the needle affect the determination of the suitable viscosity. Thus, we selected the method shown in the paper.

Q4) For Figure 3, are there diameter distribution for each sample, I suggest such data could be provided in Supporting information. More, for zeta potential value, it seems that the sample for MP1.0, MP5.0, MP10, Phy-MP10, P, and Phy-P are not stable.

A4)Thank you for your suggestion. The polydispersity index (PDI) describes the width or spread of the particle size distribution. The PDI information was shown in Figure 3.

As you questioned, the according to observed zeta potential, the particles seem to be unstable. But the particle size was stable even several days passed after the preparation at room temperature. Locating Pluronic in the outermost layer of the prepared formulation would be a reason for the stabilization of the prepared formulation. In addition, multi-peak was not observed in the present measurement.

Q5) the resolution of Figure 6 is low, please provide the improved one.

A5) Thank you for your suggestion. We used improved one. The Figure 6 may have been automagically compressed during the uploading process, resulting in low resolution of the file.

Q6) For Figure 7 and Figure 8, please improve the quality of them, such as the symbols in both captions to indicate in figures were confused and not clear, and for Figure 8b, I suggest the author separate this figure into two, then the readers could understand clearly.

Thank you for your suggestion. The symbols in Figures 7 and 8, as you pointed out, were changed. In addition, Figure 3 was also changed from with color to without color.

Q7) How about the toxicity of such formulation on other organs, were there some data?

A7) Thank you for your question. What you pointed out is very important to show the usefulness of the prepared formulation. But, in the present study, toxicity evaluation was not performed (sever toxicity was not observed). In addition, in our previous study, MGE based formulation (different formulation, but containing Pluronic and MGE) was used, and in vivo experiment over 28 days was conducted. Throughout in vivo experiment over 28 days, no severe toxicities were detected (no weight loss, no respiratory failure and no inflammation at the administrated site).

Round 2

Reviewer 3 Report

The paper was improved, I suggest the acceptance in its current form.